# Design, Synthesis, and Biological Evaluation of *N*-Acylhydrazones and Their Activity Against *Leishmania amazonensis* Promastigotes

**DOI:** 10.3390/microorganisms13071563

**Published:** 2025-07-02

**Authors:** Caio Eduardo Oliveira Monteiro, João Carlos Martins Mafra, Nubia Boechat, Edson Roberto da Silva

**Affiliations:** 1Faculdade de Zootecnia e Engenharia de Alimentos, Universidade de São Paulo, Av. Duque de Caxias Norte 225, Pirassununga 13635-900, Brazil; 2Instituto Federal do Rio de Janeiro—IFRJ, Rio de Janeiro 20270-021, Brazil; 3Laboratório de Síntese de Fármacos (LASFAR), Farmanguinhos—Fiocruz, Rio de Janeiro 21041-250, Brazil

**Keywords:** leishmaniasis, *N*-acylhydrazone-derived compounds, arginase inhibition

## Abstract

Leishmaniasis is a significant public health concern, affecting millions and causing substantial mortality, thus urgently requiring more effective and safer treatments. This study explored the potential of 33 novel *N*-acylhydrazone-derived compounds against *Leishmania amazonensis* parasites, focusing on their inhibition of the *Leishmania* arginase enzyme and promastigote growth. Compounds **8** and **18** showed over 90% inhibitory activity against promastigote cultures after 72 h of treatment. Compound **8** showed an IC_50_ of 10.5 µM (9.4–11.8 µM), while compound **18** exhibited an IC_50_ of 42.8 µM (41.3–44.4 µM). The antipromastigote effects of these compounds highlight their potential for further new drug design. These findings offer a promising starting point for addressing the pressing need for new therapeutic options against leishmaniasis. In addition, we used web-based tools to predict the compounds’ toxicity and pharmacokinetic parameters. Despite the lack of inhibition against the *L. amazonensis* arginase enzyme, further investigation into the mechanisms of action of these compounds and in vivo efficacy could contribute to the development of safer and more effective treatments for this neglected tropical disease.

## 1. Introduction

Leishmaniasis is a significant global public health concern, affecting approximately 90 countries with an annual incidence of about 1 million new cases [1]. Caused by protozoan parasites of the genus *Leishmania*, it infects both humans and animals, classifying it as an anthropozoonosis of an intracellular nature. The genus *Leishmania* comprises around 30 species, with four primarily responsible for causing disease: *Leishmania braziliensis*, *Leishmania donovani*, *Leishmania infantum*, and *Leishmania major* [2]. Visceral leishmaniasis is the most common form of the disease in Brazil, which in 2019 reported about 97% of the cases in Latin America [3]. Visceral leishmaniasis is associated with higher mortality rates if left untreated, while cutaneous leishmaniasis is generally less aggressive [4].

The disease is more prevalent in tropical and subtropical regions of both the Old and New Worlds [5]. The terms “Old World” and “New World” are historical categorizations dividing the pre-15th-century known world; the “Old World” encompasses Europe, Africa, and Asia, while the “New World” refers to the Americas [6].

Between 1988 and 2009, Brazil reported an annual average of 27,093 cases of cutaneous leishmaniasis, equating to 16 cases per 100,000 inhabitants [7]. Despite advances, treatments for leishmaniasis remain limited in efficacy and can pose significant toxicity risks to patients. Pentavalent antimonials, such as N-methylglucamine antimoniate, are currently the primary therapeutic choice for both cutaneous leishmaniasis and visceral leishmaniasis [8]. In addition to antimonials, other compounds are also used in the treatment of leishmaniasis, such as pentamidine, amphotericin B, miltefosine, and paromomycin. In all cases, treatment limitations exist, including high toxicity, low efficacy, high costs, and increased cases of drug resistance [9,10,11]. Thus, the search for more selective, less toxic, and orally active substances represents a pressing need in combating this disease [12].

*Leishmania* parasites display two distinct morphological forms: promastigotes and amastigotes. The promastigote form resides within the gut of the sandfly vector and is transmitted to humans through its bite. Once inside the human host, promastigotes transform into amastigotes, which infect macrophages, establishing these immune cells as their primary reservoir. This transformation is crucial for the parasite’s life cycle and its ability to cause disease in humans [13].

Arginase is a manganese-containing metalloenzyme that facilitates the conversion of L-arginine to L-ornithine and urea. It is found in bacteria, yeasts, plants, invertebrates, and vertebrates, and it is believed to have first emerged in bacteria [14]. In humans, arginase competes with nitric oxide synthase for the amino acid arginine. In *Leishmania*, the enzyme is synthesized and transported to the glycosome, an organelle specific to these parasites [15]. This enzyme has recently garnered significant attention for its role in various disease states, including cardiovascular system pathologies, kidney disorders, central nervous system conditions, cancer, and infectious diseases [16]. As part of efforts to develop safer and more effective antileishmanial drugs, significant work has been conducted to identify specific arginase inhibitors in *Leishmania* parasites [17].

Compounds containing the pyrazole ring have attracted interest due to their diverse biological activities, including antiparasitic [18], anti-inflammatory, and antitumor [19] properties. Furthermore, these compounds have also shown promising activity against trypanosomatids [20,21,22,23]. Bernardino et al. [24] synthesized hydrazones derived from phenyl-1H-pyrazole-4-carbohydrazides with antileishmanial activity. The two most active compounds in the series (1 and 2) exhibited EC_50_ values of 50 µM and 80 µM, respectively, against *Leishmania amazonensis* promastigotes (Figure 1). These two compounds were subsequently investigated in vivo in a murine model of cutaneous leishmaniasis, and the observed therapeutic effect was comparable to that of ketoconazole [25].

Reviriego et al. [20] demonstrated the in vitro activity of some pyrazole derivatives, substituted in positions 3 and 5, against amastigote forms of *Trypanosoma cruzi*, *Leishmania infantum*, and *Leishmania braziliensis*. The most active derivative of the series (**3**) presented an IC_50_ on the order of 7 µM for amastigote forms of *L. infantum* and *L. brasiliensis* (Figure 2) with low cytotoxicity.

These results suggest that the activity of acylhydrazones against *Leishmania* spp. parasites warrant further exploration. Consequently, a series of pyrazole *N*-acylhydrazones (compounds **4**–**36**, Figure 1) was designed to evaluate their activity against *L. amazonensis*, as well as their potential to inhibit *L. amazonensis* arginase (LaArg). This study focused on *N*-acylhydrazone derivatives as potential therapeutics for leishmaniasis. This class of compounds has been investigated for diverse therapeutic applications, including antiparasitic agents. Acylhydrazones possess significant potential for chemical and pharmaceutical modifications. The proposed mechanism of action of acylhydrazones as antiparasitic agents in *L. amazonensis* involves membrane depolarization, the production of reactive oxygen species (ROS), and alteration of cell membrane integrity [26]. Hydrazones, characterized by the azomethine -C=N-NH- group, serve not only as intermediates in the design and synthesis of various heterocyclic compounds but also stand out as valuable organic substances due to their biological effects and versatility in chemical and industrial applications. These compounds are readily synthesized, highly stable, and exhibit distinctive chemical and structural characteristics attributed to their -C=N-NH- group [27,28].

In this context, the design of derivatives **4**–**36** was undertaken with two fundamental components: the *N*-acylhydrazone nucleus, as observed in prototype **1** (highlighted in yellow, Figure 1), and the pyrazole subunit, as evidenced in substance **3** (highlighted in red, Figure 1). This approach was adopted to generate hybrid substances, with the substituents on the aromatic rings selected to elicit diverse stereo-electronic contributions. This, in turn, enabled an assessment of their influence on biological activities. Furthermore, considering the investigation of the mechanism of action, the enzyme arginase was selected as a potential target for examination through enzymatic evaluation.

## 2. Materials and Methods

### 2.1. Materials

M199 medium and penicillin/streptomycin were obtained from Life Technologies Corporation, Frederick, MD, USA. Hemin and fetal bovine serum were obtained from Sigma-Aldrich, St. Louis, MO, USA.

### 2.2. Chemistry

All reagents and solvents used were of analytical grade. The ^1^H and ^13^C nuclear magnetic resonance (NMR) spectra were obtained at 400.00 MHz using a BRUKER Avance instrument equipped with a 5 mm probe. Tetramethylsilane was used as an internal standard. The chemical shifts (*δ*) are reported in ppm, and the coupling constants (*J*) are reported in Hertz. Electron-ionization mass spectra (EI-MS, scan ES + capillary (3.0 kV)/cone (30 V)/extractor (1 V)/RF lens (1.0 V)/source temperature (150 °C)/desolvation temperature (300 °C)) were recorded using a Micromass/Waters Spectrometer (model: ZQ-4000), Waters Corporation (formerly Micromass), Milford, MA, USA. High-Resolution Mass Spectrometry (HRMS) data were obtained using an LC-MS Bruker Daltonics MicroTOF (time of flight analyzer), Bruker Daltonics is headquartered in Billerica, MA, USA, with R&D and production facilities also in Germany (Bruker Daltonics GmbH, Bremen). The melting points (mp) were determined using a Buchi model B-545 apparatus, Buchi Labortechnik AG, Flawil, Switzerland. TLC (thin layer chromatography) was performed using a silica gel F-254 glass plate (20 × 20 cm), Common suppliers include Merck KGaA, Darmstadt, Germany.

#### 2.2.1. Synthesis of 2-(1H-pyrazol-1-yl)acetohydrazide (**37a**–**d**)

A mixture of the appropriate pyrazole (0.015 mol), ethyl 2-bromoacetate (0.017 mol), K_2_CO_3_ (0.030 mol), and 30 mL of acetonitrile was stirred at reflux for 3–24 h. The mixture was cooled to 25 °C, filtered, and washed with acetonitrile (20.0 mL). The filtrate was evaporated under reduced pressure to give a yellow oil, which was dissolved in methanol and reacted with hydrazine hydrate (0.020 mol) at room temperature for 12 h. The resulting precipitate was filtered and washed with ice-cold MeOH to produce the 2-(1*H*-pyrazol-1-yl)acetohydrazide **37a**–**d** in 27–86% yield. The product obtained is suitable for the next synthetic step without purification.

2-(3,5-dimethyl-4-nitro-1*H*-pyrazol-1-yl)acetohydrazide (**37a**)

Slightly yellow solid, 86% yield, m.p. 182–184 °C. HRMS (ESI): calc. for C_7_H_11_N_5_NaO_3_^+^ 236.0760; found [M + Na] 236.0738 [err 5 ppm]. ^1^H NMR (400 MHz. DMSO-d_6_. TMS. δ in ppm): 9.42 (s; 1H; N-H); 4.78 (s; 2H; H8); 4.36 (s; 2H; NH2); 2.55 (s; 3H; H7); 2.38 (s; 3H; H6). ^13^C NMR (100 MHz, DMSO-d_6_. TMS. δ in ppm): 164.87 (C9), 144.97 (C3), 142.35 (C5), 130.35 (C4), 50.72 (C8), 13.79 (C7), 11.49 (C6).

2-(3,5-dimethyl-1*H*-pyrazol-1-yl)acetohydrazide (**37b**)

Slightly yellow solid, 27% yield, m.p. 205–207 °C. HRMS (ESI): calc. for C_7_H_13_N_4_O^+^ 169.1084; found [M + H] 169.1077 [err 3.8 ppm]. ^1^H NMR (400 MHz. DMSO-d_6_. TMS. δ in ppm): 9.23 (s; 1H; N-H); 5.78 (s; 1H; H4); 4.53 (s; 2H; H8); 4.27 (s; 2H; NH2); 2.17 (s; 3H; H7); 2.04 (s; 3H; H6). ^13^C NMR (100 MHz, DMSO-d_6_. TMS. δ in ppm): 166.35 (C9); 145.92 (C3); 139.83 (C5); 104.70 (C4); 49.79 (H8); 13.25 (C7); 10.69 (C6).

2-(1*H*-pyrazol-1-yl)acetohydrazide (**37c**)

Slightly yellow solid, 29% yield, m.p. 119–120 °C. HRMS (ESI): calc. for C_5_H_9_N_4_O^+^ 141.0771; found [M + H] 141.0764 [err 5 ppm]. ^1^H NMR (400 MHz, DMSO-d_6_. TMS. δ in ppm): 9.31 (s; 1H; N-H); 7.70 (d; J = 2.1 Hz; 1H; H5); 7.42 (d; J = 1.4 Hz; 1H; H3); 6.24 (dd; J = 2.1; 1.4 Hz; 1H; H4); 4.71 (s; 2H; H6); 4.31 (s; 2H; NH2). ^13^C NMR (100 MHz, DMSO-d_6_. TMS. δ in ppm): 165.93 (C7); 138.91 (C3); 131.27 (C5); 105.21 (C4); 52.64 (C6).

2-(4-nitro-1*H*-pyrazol-1-yl)acetohydrazide (**37d**)

Slightly yellow solid, 81% yield, m.p. 172–174 °C. HRMS (ESI): calc. For C_5_H_7_N_5_NaO_3_^+^ 208.0441; found [M + Na] 208.0441 [err −0.1 ppm]. ^1^H NMR (400 MHz, DMSO-d_6_. TMS. δ in ppm): 9.44 (s; 1H; N-H); 8.85 (d; J = 0.6 Hz; 1H; H5); 8.27 (d; J = 0.6 Hz; 1H; H3); 4.84 (s; 2H; H6); 4.37 (s; 2H; NH2). ^13^C NMR (100 MHz, DMSO-d_6_. TMS. δ in ppm): 164.64 (C7); 135.78 (C3); 134.99 (C4); 132.02 (C5); 53.50 (C6).

#### 2.2.2. Synthesis of (E)-N’-benzylidene-2-(1H-pyrazol-1-yl)acetohydrazides (**4**–**36**)

Over a mixture of the appropriate acylhydrazide **37a**–**d** (1.0 mMol = 1.0 Eq) and 15.0 mL of MeOH, a solution of the aromatic aldehyde (1.05 mMol = 1.05 Eq) in 5.0 mL of MeOH and a drop of 10% HCl solution were added. A mixture was stirred at 25 °C for 3 h. The resulting precipitate was filtered and washed with a mixture of ice-cold MeOH/H_2_O (1:1).

##### (E)-N′-benzylidene-2-(3,5-dimethyl-4-nitro-1H-pyrazol-1-yl)acetohydrazide (**4**)

Yield 83%. m.p. 219–220 °C. HRMS (ESI): calc. for C_14_H_15_N_5_NaO_3_ 324.1073; found [M + 23] 324.1065 [err 2.4 ppm]. ^1^H NMR (400 MHz, DMSO-d_6_. TMS. δ in ppm): 11.85 (s; 1H; N-H); 8.06 (s; 1H; H12); 7.73 (m; 2H; H14; H18); 7.45 (m; 3H; H15; H16; H17); 5.46 (s; 2H; H8); 2.54 (s; 3H; H7); 2.41 (s; 3H; H6). ^13^C NMR (100 MHz, DMSO-d_6_. TMS. δ in ppm): 167.13 (C9); 147.96 (C3); 144.82 (C12); 142.61 (C5); 133.79 (C4); 130.37 (C13); 130.14 (C16); 128.80 (C14; C17); 127.04 (C15; C17); 50.98 (C8); 13.79 (C7); 11.38 (C6). HPLC: 100%.

##### (E)-N′-(4-chlorobenzylidene)-2-(3,5-dimethyl-4-nitro-1H-pyrazol-1-yl)acetohydrazide (**5**)

Yield 80%. m.p. 238–240 °C. HRMS (ESI): calc. for C_14_H_14_ClN_5_NaO_3_ 358.0683; found [M + 23] 358.0675 [err −2.2 ppm]. ^1^H NMR (400 MHz, DMSO-d_6_. TMS. δ in ppm): 11.92 (s, 1H, N-H); 8.04 (s; 1H; H12); 7.77 (d; J = 8.5 Hz; 2H; H15 and H17); 7.51 (d; J = 8.5 Hz; 2H; H14 and H18); 5.46 (s; 2H; H8); 2.53 (s; 3H; H7); 2.40 (s, 3H, H6). ^13^C NMR (100 MHz, DMSO-d_6_. TMS. δ in ppm): 167.30 (C9); 146.59 (C3); 144.82 (C12); 143.28 (C5); 134.54 (C4); 132.82 (C16); 130.37 (C13); 128.87–128.81–128.69 (C14; C15; C17; C18); 51.00 (C8); 13.79 (C7); 11.38 (C6). HPLC: 100%.

##### (E)-N′-(benzo[d][1,3]dioxol-5-ylmethylene)-2-(3,5-dimethyl-4-nitro-1H-pyrazol-1-yl)acetohydrazide (**6**)

Yield 92%. m.p. 262–263 °C. HRMS (ESI): calc. for C_15_H_15_N_5_NaO_5_ 368.0971; found [M + 23] 368.0993 [err −5 ppm]. ^1^H NMR (400 MHz, DMSO-d_6_. TMS. δ in ppm): 11.73 (s; 1H; N-H); 7.95 (s; 1H; H12); 7.39 (d; J = 1.5 Hz; 1H; H14); 7.15 (dd; J = 1.5; 8 Hz; 1H; H21); 6.97 (d; J = 8 Hz; 1H; H20); 6.09 (s; 2H; H17); 5.44 (s; 2H; H8); 2.53 (s; 3H; H7); 2.41 (s; 3H; H6). ^13^C NMR (100 MHz, DMSO-d_6_. TMS. δ in ppm): 166.90 (C9); 148.99 (C19); 147.89 (C15); 144.68 (C12); 144.18 (C3); 142.45 (C5); 130.25 (C4); 128.19 (C13); 123.31 (C21); 108.27 (C20); 105.09 (C14); 101.45 (C17); 50.93 (C8); 13.69 (C7); 11.27 (C6). HPLC: 100%.

##### (E)-2-(3,5-dimethyl-4-nitro-1H-pyrazol-1-yl)-N′-(4-nitrobenzylidene)acetohydrazide (**7**)

Yield 85%. m.p. 262–263 °C. HRMS (ESI): calc. for C_14_H_14_N_6_NaO_5_ 369.0923; found [M + 23] 369.0919 [err 1.1 ppm]. ^1^H NMR (400 MHz, DMSO-d_6_. TMS. δ in ppm): 12.14 (s; 1H; N-H); 8.29 (d; 2H; *J* = 8 Hz; H15; H17); 8.16 (s; 1H; H12); 8.03 (d; 2H; *J* = 8 Hz; H14; H18); 5.51 (s; 2H; H8); 2.54 (s; 3H; H7); 2.41 (s; 3H; H6). ^13^C NMR (100 MHz, DMSO-d_6_. TMS. δ in ppm): 167.55 (C9); 147.8 (C16); 144.79 (C3); 142.55 (C12); 142.19 (C5); 139.99 (C13); 130.31 (C4); 127.94 (C14; C18); 123.88 (C15; C17); 50.92 (C8); 13.69 (C7); 11.28 (C6). HPLC: 99.7%. (Format citing only the most intense signals.)

##### (E)-2-(3,5-dimethyl-4-nitro-1H-pyrazol-1-yl)-N′-(2,4-dimethylbenzylidene)acetohydrazide (**8**)

Yield 65%. m.p. 198–199 °C. HRMS (ESI): calc. for C_16_H_19_N_5_NaO_3_ 352.1380; found [M + 23] 352.1355 [err 7 ppm]. ^1^H NMR (400 MHz, DMSO-d_6_. TMS. δ in ppm): 11.70 (s; 1H; N-H); 8.27 (s; 1H; H12); 7.69–7.65 (m; 1H; H18); 7.08 (m; 2H; H15; H17); 5.41 (s, 2H, H8); 2.54 (s, 3H, H6); 2.40 (s, 6H, H7; H19); 2.29 (s; 3H; H20). ^13^C NMR (100 MHz, DMSO-d_6_. TMS. δ in ppm): 166.77 (C9); 144.71 (C3); 143.91 (C12); 142.51 (C5); 139.39 (C14); 136.59 (C16); 131.50 (C15); 130.26 (C4); 128.93 (C17); 126.80 (C18); 126.12 (C13); 50.92 (C8); 20.78 (C19); 19.37 (C20); 13.68 (C7); 11.29 (C6). HPLC: 100%.

##### (E)-2-(3,5-dimethyl-4-nitro-1H-pyrazol-1-yl)-N′-(3-fluorobenzylidene)acetohydrazide (**9**)

Yield 87%. m.p. 221–223 °C. HRMS (ESI): calc. for C_14_H_14_FN_5_NaO_3_ 342.0973; found [M + 23] 342.0958 [err 4.2 ppm]. ^1^H NMR (400 MHz, DMSO-d_6_. TMS. δ in ppm): 11.94 (s; 1H; N-H); 8.05 (s; 1H; H12); 7.65–7.61 (m; 1H; H18); 7.58–7.56 (m; 1H; H15); 7.52–7.47 (m; 1H; H14); 7.30–7.25 (m; 1H; H16); 5.48 (s; 2H; H8); 2.53 (s; 3H; H6); 2.41 (s; 3H; H7). 167.29 (C9); 162.32 (d; J = 242.2 Hz; C17); 144.73 (C3); 143.08 (d; J = 2.8 Hz; C12); 142.48 (C5); 136.30 (d; J = 8 Hz; C13); 130.76 (d; J = 8.3 Hz; C15); 130.28 (C4); 123.59 (d; J = 2.4 Hz; C14); 116.8 (d; J = 21.4 Hz; C16); 112.8 (d; J = 22.5 Hz; C18); 50.95 (C8); 13.69 (C7); 11.26 (C6). HPLC: 100%.

##### (E)-N′-(3-bromobenzylidene)-2-(3,5-dimethyl-4-nitro-1H-pyrazol-1-yl)acetohydrazide (**10**)

Yield 84%. m.p. 229–230 °C. HRMS (ESI): calc. for C_14_H_14_BrN_5_NaO_3_ 402.0172; found [M + 23] 402.0158 [err 3.5 ppm]. ^1^H NMR (400 MHz, DMSO-d_6_. TMS. δ in ppm): 11.94 (s; 1H; N-H); 8.02 (s; 1H; H12); 7.99 (m; 1H; H14); 7.73–7.71 (m; 1H; H18); 7.63–7.61 (m; 1H; H16); 7.43–7.39 (m; 1H; H17); 5.48 (s; 2H; H8); 2.53 (s; 3H; H6); 2.41 (s; 6H; H7). ^13^C NMR (100 MHz, DMSO-d_6_. TMS. δ in ppm): 167.38 (C9); 144.80 (C3); 142.90 (C12); 142.57 (C5); 136.26 (C13); 132.65 (C16); 130.91 (C14); 130.35 (C4); 128.99 (C17); 126.42 (C18); 122.23 (C15); 51.07 (C8); 13.78 (C7); 11.35 (C6). HPLC: 99.8%.

##### (E)-2-(3,5-dimethyl-4-nitro-1H-pyrazol-1-yl)-N′-(2-fluorobenzylidene)acetohydrazide (**11**)

Yield 68%. m.p. 205–207 °C. HRMS (ESI): calc. for C_14_H_14_FN_5_NaO_3_ 342.0973; found [M + 23] 342.0971 [err 0.7 ppm]. ^1^H NMR (400 MHz, DMSO-d_6_. TMS. δ in ppm): 11.96 (s; 1H; N-H); 8.26 (s; 1H; H12); 8.02–7.97 (m; 1H; H18); 7.53–7.48 (m; 1H; H16); 7.33–7.28 (m; 2H; H15; H17); 5.48 (s; 2H; H8); 2.54 (s; 3H; H6); 2.41 (s; 3H; H7). ^13^C NMR (100 MHz, DMSO-d_6_. TMS. δ in ppm): 160.62 (d; J = 248.5 Hz; C14); 145.75 (C3); 142.53 (C5); 137.39 (d; J = 4.5 Hz; C12); 132.02 (d; J = 8.5 Hz; C16); 130.29 (C4); 126.49 (d; J = 2.2 Hz; C17); 124.77 (d; J = 3.1 Hz; C18); 121.28 (d; J = 9.9 Hz; C13); 115.9 (d; J = 20.6 Hz; C15); 50.87 (C8); 13.69 (C7); 11.28 (C6). HPLC: 99.8%.

##### (E)-2-(3,5-dimethyl-4-nitro-1H-pyrazol-1-yl)-N′-(2-methoxybenzylidene)acetohydrazide (**12**)

Yield 95%. m.p. 254–256 °C. HRMS (ESI): calc. for C_15_H_17_N_5_NaO_4_ 354.1173; found [M + 23] 354.1174 [err −0.3 ppm]. ^1^H NMR (400 MHz, DMSO-d_6_. TMS. δ in ppm): 11.78 (s; 1H; N-H); 8.39 (s; 1H; H12); 7.90–7.88 (m; 1H; H18); 7.49–7.41 (m; 1H; H16); 7.12–7.10 (m; 1H; H17); 7.03–6.99 (m; 1H; H15); 5.44 (s; 2H; H8); 3.86 (s; 3H; H19); 2.53 (s; 3H; H6); 2.41 (s; 6H; H7). ^13^C NMR (100 MHz, DMSO-d_6_. TMS. δ in ppm): 166.9 (C9); 157.61 (C14); 144.71 (C3); 142.49 (C12); 140.17 (C5); 131.57 (C16); 130.26 (C4); 125.53 (C18); 121.69 (C17); 120.57 (C13); 111.73 (C15); 55.61 (C19); 50.93 (C8); 13.69 (C7); 11.29 (C6). HPLC: 99.8%.

##### (E)-2-(3,5-dimethyl-4-nitro-1H-pyrazol-1-yl)-N′-(4-fluorobenzylidene)acetohydrazide (**13**)

Yield 77%. m.p. 222–223 °C. HRMS (ESI): calc. for C_14_H_14_FN_5_NaO_3_ 342.0973; found [M + 23] 342.0972 [err 0.2 ppm]. ^1^H NMR (400 MHz, DMSO-d_6_. TMS. δ in ppm): 11.85 (s; 1H; N-H); 8.05 (s; 1H; H12); 7.83–7.76 (m; 2H; H14; H18); 7.32–7.27 (m; 2H; H15; H17); 5.46 (s; 2H; H8); 2.53 (s; 3H; H6); 2.41 (s; 3H; H7). ^13^C NMR (100 MHz, DMSO-d_6_. TMS. 167.06 (C9); 163.03 (d; J = 246.3 Hz; C16); 146.76 (C3); 144.72 (C12); 143.37 (C5); 142.50 (C4); 130.37–130.34–130.28 (ms; C13); 129.17 (d; J = 8.5 Hz; C14 and C18); 115.77 (d; J = 21.8 Hz; C15 and C17); 50.89 (C8); 13.69 (C7); 11.28 (C6). HPLC: 99.8%.

##### (E)-N′-(3-chlorobenzylidene)-2-(3,5-dimethyl-4-nitro-1H-pyrazol-1-yl)acetohydrazide (**14**)

Yield 88%. m.p. 218–219 °C. HRMS (ESI): calc. for C_14_H_14_ClN_5_NaO_3_ 358.0678; found [M + 23] 358.0673 [err 1.2 ppm]. ^1^H NMR (400 MHz, DMSO-d_6_. TMS. δ in ppm): 11.95 (s; 1H; N-H); 8.03 (s; 1H; H12); 7.83 (m; 1H; H14); 7.70–7.68 (m; 1H; H18); 7.50–7.47 (m; 2H; H16; H17); 5.49 (s; 2H; H8); 2.53 (s; 3H; H6); 2.41 (s; 6H; H7). ^13^C NMR (100 MHz, DMSO-d_6_. TMS. δ in ppm): 167.30 (C9); 144.72 (C3); 142.89 (C12); 142.48 (C5); 135.96 (C13); 133.59 (C16); 130.57 (C14); 130.27 (C4); 129.67 (C17); 126.04 (C18); 125.96 (C15); 50.98 (C8); 13.69 (C7); 11.26 (C6). HPLC: 99.8%.

##### (E)-2-(3,5-dimethyl-1H-pyrazol-1-yl)-N′-(3-hydroxybenzylidene)acetohydrazide (**15**)

Yield 89%. m.p. 209–210 °C. HRMS (ESI): calc. for C_14_H_17_N_4_NaO_2_ 295.1165; found [M + 23] 295.1168 [err −0.9 ppm]. ^1^H NMR (400 MHz, DMSO-d_6_. TMS. δ in ppm): 11.57 (s; 1H; N-H); 9.61 (a; 1H; O-H); 7.94 (s; 1H; H12); 7.26–7.22 (m; 1H; H18); 7.16–7.03 (m; 2H; H14; H17); 6.84–6.82 (m; 1H; H16); 5.82 (s; 1H; H4); 5.19 (s; 2H; H8); 2.15 (s; 3H; H6); 2.07 (s; 6H; H7). ^13^C NMR (100 MHz, DMSO-d_6_. TMS. δ in ppm): 168.36 (C9); 157.53 (C15); 147.35 (C3); 145.73 (C12); 144.02 (C5); 135.09 (C13); 129.75 (C17); 118.26 (C18); 117.17 (C16); 112.67 (C14); 104.64 (C4); 49.32 (C8); 13.16 (C7); 10.50 (C6). HPLC: 99.8%.

##### (E)-N′-(3-chlorobenzylidene)-2-(3,5-dimethyl-1H-pyrazol-1-yl)acetohydrazide (**16**)

Yield 74%. m.p. 214–215 °C. HRMS (ESI): calc. for C_14_H_16_ClN_4_O 291.1007; found [M + 1] 291.1008 [err −0.3 ppm]. ^1^H NMR (400 MHz, DMSO-d_6_. TMS. δ in ppm): 11.74 (s; 1H; N-H); 8.01 (s; 1H; H12); 7.82 (s;1H; H14); 7.68–7.66 (m; 1H; H18); 7.48–7.47 (m; 2 H; H16; H17); 5.82 (s; 1H; H4); 5.23 (s; 2H; H8); 2.15 (s; 3H; H7); 2.08 (s; 3H; H6). ^13^C NMR (100 MHz, DMSO-d_6_. TMS. δ in ppm): 168.71 (C9); 145.72 (C3); 142.18 (C12); 139.93 (C5); 136.12 (C13); 133.57 (C15); 130.57 (C16); 129.50 (C17); 125.97 (C14); 125.76 (C18); 104.61 (C4); 49.51 (C8); 13.18 (C7); 10.48 (C6). HPLC: 100%.

##### (E)-N′-(benzo[d][1,3]dioxol-5-ylmethylene)-2-(3,5-dimethyl-1H-pyrazol-1-yl)acetohydrazide (**17**)

Yield 70%. m.p. 214–216 °C. HRMS (ESI): calc. for C_15_H_17_N_4_O_3_ 301.1295; found [M + 1] 301.1289 [err 1.9 ppm]. ^1^H NMR (400 MHz, DMSO-d_6_. TMS. δ in ppm): 11.52 (s; 1H; N-H); 7.92 (s; 1H; H12); 7.35 (d; J = 1.5 Hz; 1H; H14); 7.13 (dd; J = 1.5; 8 Hz; 1H; H21); 6.98 (d; J = 8 Hz; 1H; H20); 6.08 (s; 2H; H17); 5.82 (s; 1H; H4); 5.19 (s; 2H; H8); 2.14 (s; 3H; H7); 2.07 (s; 3H; H6). ^13^C NMR (100 MHz, DMSO-d_6_. TMS. δ in ppm): 168.45 (C9); 148.95 (C19); 147.97 (C15); 145.75 (C3); 143.59 (C12); 139.99 (C5); 128.44 (C13); 123.19 (C21); 108.38 (C20); 105.09 (C14); 104.68 (C4); 101.51 (C17); 49.57 (C8); 13.28 (C7); 10.60 (C6). HPLC: 99.5%.

##### (E)-2-(3,5-dimethyl-1H-pyrazol-1-yl)-N′-(2-hydroxybenzylidene)acetohydrazide (**18**)

Yield 90%. m.p. 175–177 °C. HRMS (ESI): calc. for C_14_H_17_N_4_O_2_ 273.1346; found [M + 1] 273.1349 [err −0.9 ppm]. ^1^H NMR (400 MHz, DMSO-d_6_. TMS. δ in ppm): 11.93 (s; 1H; N-H); 10.97 (s; 1H; O-H); 8.33 (s; 1H; H12); 7.74–7.22 (m; 1H; H18); 7.31–7.23 (m; 1H; H16); 6.92–6.85 (m, 2H, H15; H17); 5.83 (s; 1H; H4); 5.18 (s; 2H; H8); 2.15 (s; 3H; H7); 2.08 (s; 3 H; H6). ^13^C NMR (100 MHz, DMSO-d_6_. TMS. δ in ppm): 168.23 (C9); 157.29 (C14); 147.61 (C3); 146.31 (C12); 141.34 (C5); 131.23 (C16); 129.19 (C18); 120.06 (C17); 119.36 (C15); 116.35 (C13); 104.87 (C4); 50.19 (C8); 13.26 (C7); 10.60 (C6). HPLC: 94%.

##### (E)-N′-(2,4-dichlorobenzylidene)-2-(3,5-dimethyl-1H-pyrazol-1-yl)acetohydrazi-de (**19**)

Yield 76%. m.p. 210–212 °C. HRMS (ESI): calc. for C_14_H_14_Cl_2_N_4_NaO 347.0437; found [M + 23] 347.0453 [err −4.8 ppm]. ^1^H NMR (400 MHz, DMSO-d_6_. TMS. δ in ppm): 11.85 (s; 1H; N-H); 8.35 (s; 1H; H12); 8.05 (d; *J* = 8 Hz; 1H; H18); 7.72 (d; *J* = 2 Hz; 1H; H15); 7.51 (dd; *J* = 2; 8 Hz; H17); 5.82 (s; 1H; H4); 5.23 (s; 2H; H8); 2.14 (s; 3H; H7); 2.07 (s; 3H; H6). ^13^C NMR (100 MHz, DMSO-d_6_. TMS. δ in ppm): 168.73 (C9); 145.78 (C3); 139.96 (C5); 138.81 (C12); 134.90 (C13); 133.56 (C14); 130.26 (C16); 129.26 (C15); 128.15–127.99–127.84 (C17; C18); 104.65 (C4); 49.46 (C8); 13.17 (C7); 10.48 (C6). HPLC: 99.8%.

##### (E)-2-(3,5-dimethyl-1H-pyrazol-1-yl)-N′-(4-nitrobenzylidene)acetohydrazide (**20**)

Yield 97%. m.p. 212–214 °C. HRMS (ESI): calc. for C_14_H_15_N_5_NaO_3_ 324.1067; found [M + 23] 324.1079 [err −3.8 ppm]. ^1^H NMR (400 MHz, DMSO-d_6_. TMS. δ in ppm): 11.92 (s; 1H; N-H); 8.34–8.27 (m; 2H; H15; H17); 8.13 (s; 1H; H12); 8.01–7.96 (m; 2H; H14; H18); 5.83 (s; 2H; H8); 2.15 (s; 3H; H7); 2.08 (s; 3H; H6). ^13^C NMR (100 MHz, DMSO-d_6_. TMS. δ in ppm): 169.05 (C9); 147.79 (C16); 145.94 (C3); 141.60 (C12); 140.27 (C5); 140.11 (C13); 127.91 (C14; C18); 124.01 (C15; C17); 104.79 (C4); 49.55 (C8); 13.28 (C7); 11.59 (C6). HPLC: 99.7%.

##### (E)-N′-(4-chlorobenzylidene)-2-(3,5-dimethyl-1H-pyrazol-1-yl)acetohydrazide (**21**)

Yield 74%. m.p. 183–185 °C. HRMS (ESI): calc. for C_14_H_15_ClN_4_NaO 313.0827; found [M + 23] 313.0835 [err −2.8 ppm]. ^1^H NMR (400 MHz, DMSO-d_6_. TMS. δ in ppm): 11.69 (s; 1H; N-H); 8.02 (s; 1H; H12); 7.75 (d; J = 8.5 Hz; 2H; H15; H17); 7.51 (d; J = 8.5 Hz; 2H; H14; H18); 5.82 (s; 1H; H4); 5.21 (s; 2H; H8); 2.15 (s; 3H; H7); 2.07 (s; 3H; H6). ^13^C NMR (100 MHz, DMSO-d_6_. TMS. δ in ppm): 168.67 (C9); 145.84 (C3); 142.64 (C12); 140.07 (C5); 134.40 (C16); 132.93 (C13); 128.89 (C15; C17); 128.59 (C14; C18); 104.73 (C4); 49.51 (C8); 13.28 (C7); 10.59 (C6). HPLC: 100%.

##### (E)-2-(3,5-dimethyl-1H-pyrazol-1-yl)-N′-(3-nitrobenzylidene)acetohydrazide (**22**)

Yield 83%. m.p. 209–210 °C. HRMS (ESI): calc. for C_14_H_15_N_5_NaO_3_ 324.1067; found [M + 23] 324.1076 [err −2.9 ppm]. ^1^H NMR (400 MHz, DMSO-d_6_. TMS. δ in ppm): 11.87 (s; 1H; N-H); 8.53 (m; 1H; H14); 8.15 (s; 1H; H12); 8.27–8.24 (m; 1H; H16); 8.20–8.14 (m; 1H; H18); 7.76–7.72 (m; 1H; H17); 5.83 (s; 1H; H4); 5.26 (s; 2H; H8); 2.16 (s; 3H; H7); 2.08 (s; 3H; H6). ^13^C NMR (100 MHz, DMSO-d_6_. TMS. δ in ppm): 168.80 (C9); 148.15 (C15); 145.8 (C3); 141.59 (C12); 140.01 (C5); 135.75 (C13); 132.97 (C18); 130.29 (C17); 124.09 (C16); 121.0 (C14); 104.66 (C4); 49.51 (C8); 13.17 (C7); 10.49 (C6). HPLC: 98.4%.

##### (E)-2-(3,5-dimethyl-1H-pyrazol-1-yl)-N′-(2-nitrobenzylidene)acetohydrazide (**23**)

Yield 62%. m.p. 232–234 °C. HRMS (ESI): calc. for C_14_H_15_N_5_NaO_3_ 324.1067; found [M + 23] 324.1074 [err −2.2 ppm]. ^1^H NMR (400 MHz, DMSO-d_6_. TMS. δ in ppm): 11.91 (s; 1H; N-H); 8.40 (s; 1H; H12); 8.17–8.10 (m; 1H; H18); 8.08–8.02 (m; 1H; H15); 7.83–7.78 (m; 1H; H17); 7.71–7.65 (m; 1H; H16); 5.83 (s; 1H; H4); 5.20 (s; 2H; H8); 2.15 (s; 3H; H7); 2.08 (s; 3H; H6). ^13^C NMR (100 MHz, DMSO-d_6_. TMS. δ in ppm): 168.92 (C9); 148.04 (C14); 145.92 (C3); 143.18 (C12); 140.08 (C5); 139.47 (C17); 133.54 (C16); 130.61 (C18); 128.53 (C13); 124.54 (C15); 104.80 (C4); 49.56 (C8); 13.28 (C7); 10.58 (C6). HPLC: 95.6%.

##### (E)-N′-(2,4-difluorobenzylidene)-2-(3,5-dimethyl-1H-pyrazol-1-yl)acetohydrazide (**24**)

Yield 91%. m.p. 209–210 °C. HRMS (ESI): calc. for C_14_H_14_F_2_N_4_NaO 315.1028; found [M + 23] 315.1035 [err −2.2 ppm]. ^1^H NMR (400 MHz, DMSO-d_6_. TMS. 11.74 (s; 1H; N-H); 8.17 (s; 1H; H12); 8.05–7.99 (m; 1H; H18); 7.40–7.34 (m; 1H; H17); 7.23–7.18 (m; 1H; H15); 5.82 (s; 1H; H4); 5.21 (s; 2H; H8); 2.14 (s; 3H; H6); 2.07 (s; 3H; H7). ^13^C NMR (100 MHz, DMSO-d_6_. TMS. δ in ppm): 168.61 (C9); 163.06 (dd; J = 251.7 and 12.5 Hz; C14); 160.73 (dd; J = 251 and 12.5 Hz; C16); 145.76 (C12); 139.96 (C3); 135.91 (C5); 128.01 (dd; J = 9.7 and 4.1 Hz; C18); 118.33 (dd; J = 10.1 and 3.8 Hz; C13); 112.48 (dd; J = 21.8 and 3.3 Hz; C17); 104.77 (C4); 104.39 (dd; J = 26 and 25 Hz; C15); 49.40 (C8); 13.17 (C7); 10.48 (C6). HPLC: 99.8%.

##### (E)-N′-(3-bromobenzylidene)-2-(3,5-dimethyl-1H-pyrazol-1-yl)acetohydrazide (**25**)

Yield 87%. m.p. 220–222 °C. HRMS (ESI): calc. for C_14_H_15_BrN_4_NaO 357.0321; found [M + 23] 357.0326 [err −1.3 ppm]. ^1^H NMR (400 MHz, DMSO-d_6_. TMS. δ in ppm): 11.73 (s; 1H; N-H); 7.99 (s; 1H; H12); 7.95 (m; 1H; H14); 7.72–7.70 (m; 1H; H18); 7.63–7.60 (m; 1H; H16); 7.43–7.38 (m; 1H; H17); 5.82 (s; 1H; H4); 5.23 (s; 2H; H8); 2.15 (s; 3H; H7); 2.08 (s; 3H; H6). ^13^C NMR (100 MHz, DMSO-d_6_. TMS. δ in ppm): 168.79 (C9); 145.82 (C3); 142.20 (C12); 140.02 (C5); 136.43 (C13); 132.49 (C16); 130.92 (C14); 129.94 (C17); 126.22 (C18); 122.21 (C15); 104.70 (C4); 49.61 (C8); 13.28 (C7); 10.58 (C6). HPLC: 99.6%.

##### (E)-2-(3,5-dimethyl-1H-pyrazol-1-yl)-N′-(2,4-dinitrobenzylidene)acetohydrazide (**26**)

Yield 94%. m.p. 205–207 °C. HRMS (ESI): calc. for C_14_H_14_N_6_NaO_5_ 369.0918; found [M + 23] 369.0930 [err −3.3 ppm]. ^1^H NMR (400 MHz, DMSO-d_6_. TMS. δ in ppm): 12.17 (s; 1H; N-H); 8.79 (d; J = 2 Hz; 1H; H15); 8.53 (dd; J = 8.8 and 2 Hz; 1H; H17); 8.46 (s; 1H; H12); 8.41 (d; J = 8.8 Hz; 1H; H18); 5.83 (s; 1H; H4); 5.25 (s; 2H; H8); 2.15 (s; 3H; H6); 2.08 (s; 3H; H7). ^13^C NMR (100 MHz, DMSO-d_6_. TMS. δ in ppm): 169.14 (C9); 147.50 (C16); 147.16 (C14); 145.91 (C3); 140.02 (C12); 137.85 (C5); 133.66 (C13); 129.79 (C18); 127.30 (C17); 120.16 (C15); 104.75 (C4); 49.47 (C8); 13.17 (C7); 10.46 (C6). HPLC: 98.7%.

##### (E)-2-(3,5-dimethyl-1H-pyrazol-1-yl)-N′-(2,4-dimethylbenzylidene)acetohydrazide (**27**)

Yield 77%. m.p. 195–196 °C. HRMS (ESI): calc. for C_16_H_20_N_4_NaO 307.1529; found [M + 23] 307.1525 [err 1.5 ppm]. ^1^H NMR (400 MHz, DMSO-d_6_. TMS. δ in ppm): 11.49 (s; 1H; N-H); 8.24 (s; 1H; H12); 7.67–7.65 (m; 1H; H18); 7.09–7.07 (m; 2H; H15; H17); 5.82 (s; 1H; H4); 5.18 (s; 2H; H8); 2.40 (s; 3H; H6); 2.29 (s; 3H; H19); 2.15 (s; 3H; H7); 2.07 (C20). ^13^C NMR (100 MHz, DMSO-d_6_. TMS. δ in ppm): 168.34 (C9); 145.79 (C3); 143.34 (C12); 140.06 (C5); 139.28 (C14); 136.55 (C16); 131.6 (C15); 129.19 (C18); 126.91 (C17); 126.79 (13); 104.72 (C4); 49.6 (C8); 20.87 (C19); 19.51 (C20); 13.27 (C7); 10.6 (C6). HPLC: 100%.

##### (E)-N′-(4-chlorobenzylidene)-2-(1H-pyrazol-1-yl)acetohydrazide (**28**)

Yield 95%. m.p. 228–230 °C. HRMS (ESI): calc. for C_12_H_11_ClN_4_NaO 285.0514; found [M + 23] 285.0519 [err −1.8 ppm]. ^1^H NMR (400 MHz, DMSO-d_6_. TMS. δ in ppm): 11.72 (s; 1H; N-H); 8.02 (s; 1H; H10); 7.77–7.72 (m; 3H; H5; H13; H15); 7.52–7.50 (m; 2H; H12; H16); 7.47–7.45 (m; 1H; H3); 6.28–6.26 (m; 1H; H4); 5.40 (s; 2H; H6). ^13^C NMR (100 MHz, DMSO-d_6_. TMS. δ in ppm): 168.58 (C7); 142.50 (C10); 138.60 (C3); 134.30 (C14); 132.81 (C11); 131.54 (C5); 128.79 (C13; C15); 128.49 (C12; C16); 105.15 (C4); 52.08 (C6). HPLC: 100%.

##### (E)-N′-(2-methoxybenzylidene)-2-(1H-pyrazol-1-yl)acetohydrazide (**29**)

Yield 62%. m.p. 218–220 °C. HRMS (ESI): calc. for C_13_H_14_N_4_NaO_2_ 281.1009; found [M + 23] 281.1021 [err 4.3 ppm]. ^1^H NMR (400 MHz, DMSO-d_6_. TMS. δ in ppm): 11.62 (s; 1H; N-H); 8.36 (s; 1H; H10); 7.89–7.88 (m; 1H; H-16); 7.74–7.73 (m; 1H; H5); 7.47–7.39 (m; 2H; H3; H14); 7.11–7.09 (m; 1H; H13); 7.03–6.99 (m; 1H; H15); 6.28–6.26 (m; 1H; H4); 5.38 (s; 2H; H6); 3.86 (s; 3H; H17). ^13^C NMR (100 MHz, DMSO-d_6_. TMS. δ in ppm): 168.35 (C7); 157.52 (C12); 139.40 (C10); 138.56 (C3); 131.53 (C14); 131.38 (C5); 125.37 (C16); 121.82 (C15); 120.62 (C11); 111.69 (C13); 105.11 (C4); 55.57 (C17); 52.11 (C6). HPLC: 98.5%.

##### (E)-N′-(3-fluorobenzylidene)-2-(4-nitro-1H-pyrazol-1-yl)acetohydrazide (**30**)

Yield 78%. m.p. 186–187 °C. HRMS (ESI): calc. for C_12_H_10_FN_5_NaO_3_ 314.0660; found [M + 23] 314.0649 [err 3.4 ppm]. ^1^H NMR (400 MHz, DMSO-d_6_. TMS. δ in ppm): 11.95 (s; 1H; N-H); 8.89 (s; 1H; H5); 8.30 (s; 1H; H3); 8.04 (s; 1H; H10); 7.64–7.61 (m; 1H; H16); 7.57–7.55 (m; 1H; H13); 7.52–7.47 (m; 1H; H12); 7.30–7.25 (m; 1H; H14); 5.54 (s; 2H; H6). ^13^C NMR (100 MHz, DMSO-d_6_. TMS. δ in ppm): 167.61 (C7); 162.44 (d; J = 242.2 Hz; C15); 143.16 (d; J = 2.87 Hz; C10); 136.37 (d; J = 7.9 Hz; C11); 135.55 (C3); 135.14 (C4); 132.29 (C5); 130.9 (d; J = 8.4 Hz; C13); 123.66 (d; J = 2.4 Hz; C12); 116.9 (d; J = 21.4 Hz; C14); 112.9 (d; J = 22.5 Hz; C16); 53.82 (C6). HPLC: 100%.

##### (E)-2-(4-nitro-1H-pyrazol-1-yl)-N′-(3-nitrobenzylidene)acetohydrazide (**31**)

Yield 78%. m.p. 209–211 °C. HRMS (ESI): calc. for C_12_H_10_N_6_NaO_5_ 341.0605; found [M + 23] 341.0598 [err 2.1 ppm]. ^1^H NMR (400 MHz, DMSO-d_6_. TMS. δ in ppm): 12.09 (s; 1H; N-H); 8.89 (d; J = 0.6 Hz; 1H; H5); 8.56–8.55 (m; 1H; H16); 8.31 (d; J = 0.6 Hz; 1H; H3); 8.29–8.25 (m; 1H; H14); 8.23–8.21 (m; 1H; H12); 8.18 (s; 1H; H10); 7.77–7.73 (m; 1H; H13); 5.58 (s; 2H; H6). ^13^C NMR (100 MHz, DMSO-d_6_. TMS. δ in ppm): 167.62 (C7); 148.17 (C15); 142.26 (C10); 135.56 (C3); 135.46 (C4); 135.04 (C11); 133.0 (C5); 132.18 (C12); 130.30 (C13); 124.27 (C14); 121.18 (C16); 53.70 (C6). HPLC: 98.3%.

##### (E)-N′-(4-chlorobenzylidene)-2-(4-nitro-1H-pyrazol-1-yl)acetohydrazide (**32**)

Yield 83%. m.p. 217–219 °C. HRMS (ESI): calc. for C_12_H_10_ClN_5_NaO_3_ 330.0364; found [M + 23] 330.0348 [err 5.1 ppm]. ^1^H NMR (400 MHz, DMSO-d_6_. TMS. δ in ppm): 11.92 (s; 1H; N-H); 8.89 (d; J = 0.5 Hz; 1H; H5); 8.30 (d; J = 0.5 Hz; 1H; H3); 8.04 (s; 1H; H10); 7.79–7.76 (m; 2H; H12; H16); 7.52–7.50 (m; 2H; H13; H15); 5.52 (s; 2H; H6). ^13^C NMR (100 MHz, DMSO-d_6_. TMS. δ in ppm): 167.36 (C7); 143.19 (C10); 135.44 (C3); 135.03 (C14); 134.49 (C4); 132.65 (C5); 132.18 (C11); 128.84–128.81–128.77 (ms; C12; C16); 128.59 (C13; C15); 53.66 (C6). HPLC: 100%.

##### (E)-2-(4-nitro-1H-pyrazol-1-yl)-N′-(4-nitrobenzylidene)acetohydrazide (**33**)

Yield 85%. m.p. 226–228 °C. HRMS (ESI): calc. for C_12_H_10_N_6_NaO_5_ 341.0605; found [M + 23] 341.0605 [err −0.1 ppm]. ^1^H NMR (400 MHz, DMSO-d_6_. TMS. δ in ppm): 12.16 (s; 1H; N-H); 8.89 (d; J = 0.5 Hz; 1H; H5); 8.31 (d; J = 0.5 Hz; 1H; H3); 8.29–8.27 (m; 2H; H13; H15); 8.15 (s; 1H; H10); 8.04–8.01 (m; 2H; H12; H16); 5.57 (s; 2H; H6). ^13^C NMR (100 MHz, DMSO-d_6_. TMS. δ in ppm): 167.75 (C7); 147.8 (C14); 142.16 (C10); 139.95 (C11); 135.49 (C3); 135.08 (C4); 132.19 (C5); 127.92 (C12; C16); 123.9 (C13; C15); 53.68 (C6). HPLC: 99.6%.

##### (E)-N′-(3-bromobenzylidene)-2-(4-nitro-1H-pyrazol-1-yl)acetohydrazide (**34**)

Yield 82%. m.p. 205–207 °C. HRMS (ESI): calc. for C_12_H_10_BrN_5_NaO_3_ 373.9859; found [M + 23] 373.9851 [err 2.2 ppm]. ^1^H NMR (400 MHz, DMSO-d_6_. TMS. δ in ppm): 11.96 (s; 1H; N-H); 8.89 (d; J = 0.4 Hz; 1H; H5); 8.30 (d; J = 0.4 Hz; 1H; H3); 8.02 (s; 1H; H10); 7.99–7.98 (m; 1H; H16); 7.74–7.72 (m; 1H; H12); 7.64–7.61 (m; 1H; H14); 7.44–7.39 (m; 1H; H13); 5.54 (s; 2H; H6). ^13^C NMR (100 MHz, DMSO-d_6_. TMS. δ in ppm): 167.62 (C7); 142.88 (C10); 136.26 (C3); 135.53 (C4); 135.12 (C11); 132.68(C5); 132.27 (C14); 130.95 (C16); 129.04 (C13); 126.35 (C12); 122.26 (C15); 53.84 (C6). HPLC: 98.1%.

##### (E)-N′-(2-fluorobenzylidene)-2-(4-nitro-1H-pyrazol-1-yl)acetohydrazide (**35**)

Yield 76%. m.p. 200–202 °C. HRMS (ESI): calc. for C_12_H_10_FN_5_NaO_3_ 314.0660; found [M + 23] 314.0658 [err 0.6 ppm]. ^1^H NMR (400 MHz, DMSO-d_6_. TMS. δ in ppm): 11.97 (s; 1H; N-H); 8.89 (s; 1H; H5); 8.31 (s; 1H; H3); 8.25 (s; 1H; H10); 8.01–7.97 (m; 1H; H16); 7.53–7.48 (m; 1H; H14); 7.33–7.28 (m; 2H; H13; H15); 5.53 (s; 2H; H6). ^13^C NMR (100 MHz, DMSO-d_6_. TMS. δ in ppm): 167.49 (C7); 160.72 (d; J = 248.6 Hz; C12); 137.43 (d; J = 4.6 Hz; C10); 135.56 (C4); 135.15 (C3); 132.29 (C5); 132.13 (d; J = 8.4 Hz; C14); 126.51 (d; J = 2.3 Hz; C15); 124.92 (d; J = 3.2 Hz; C16); 121.34 (d; J = 9.7 Hz; C11); 116.05 (d; J = 20.7 Hz; C13); 53.74 (C6). HPLC: 100%.

##### (E)-2-(4-nitro-1H-pyrazol-1-yl)-N′-(2-nitrobenzylidene)acetohydrazide (**36**)

Yield 89%. m.p. 198–200 °C. HRMS (ESI): calc. for C_12_H_10_N_6_NaO_5_ 341.0605; found [M + 23] 341.0602 [err 0.9 ppm]. ^1^H NMR (400 MHz, DMSO-d_6_. TMS. δ in ppm): 12.13 (s; 1H; N-H); 8.89 (s; 1H; H5); 8.43 (s; 1H; H3); 8.31 (m; 1H; H10); 8.17–8.14 (m; 1H; H16); 8.1–8.06 (m; 1H; H13); 7.83–7.76 (m; 1H; H15); 7.72–7.67 (m; 1H; H14); 5.52 (s; 2H; H6). ^13^C NMR (100 MHz, DMSO-d_6_. TMS. δ in ppm): 167.61 (C7); 147.97 (C12); 140.14 (C10); 135.47 (C3); 135.03 (C4); 133.48 (C15); 132.19 (C5); 130.62 (C14); 128.4 (C16); 127.89 (C11); 124.48 (C13); 53.64 (C6). HPLC: 98.7%.

##### Synthesis of (E)-2-(3,5-dimethyl-1H-pyrazol-1-yl)-N-methyl-N′-(3-nitrobenzy-lidene)acetohydrazide (**37**)

A mixture of the appropriate compound **22** (0.30 mmol), methyl iodide (0.91 mmol), K_2_CO_3_ (0.91 mmol), and 10 mL of acetone was stirred and allowed to reflux for 12h. The mixture was evaporated under reduced pressure. The resulting precipitate was treated with sodium bisulfite solution, filtered, and washed with ice-cold MeOH/H_2_O (1:1).

Yield 81%. m.p. °C. HRMS (ESI): calc. for C_15_H_17_N_5_NaO_3_ 338.1224; found [M + 23] 338.1218 [err 1.8 ppm]. ^1^H NMR (400 MHz, DMSO-d_6_. TMS. δ in ppm): 8.60 (s; 1H; H14); 8.27 (m; 2H; H16; H18); 8.22 (s; 1H; H12); 7.76 (m; 1H; H17); 5.84 (s; 1H; H4); 5.42 (s; 2H; H8); 3.36 (s; 3H; H19); 2.14 (s; 3H; H7); 2.08 (s; 3H; H6). ^13^C NMR (100 MHz, DMSO-d_6_. TMS. δ in ppm): 168.42 (C9); 148.25 (C15); 145.9 (C3); 140.11 (C12); 139.24 (C5); 136.39 (C13); 132.88 (C18); 130.39 (C17); 124.09 (C16); 121.71 (C14); 104.8 (C4); 50.49 (C8); 28.48 (C19); 13.27 (C7); 10.57 (C6) HPLC: 94.8%.

### 2.3. Screening and IC_50_ of N-Acylhydrazone Derivatives in Promastigote Culture

*Leishmania amazonensis* promastigotes were maintained at 25 °C in M199 medium supplemented with 10% fetal bovine serum, 50 U/mL of penicillin, 50 µg/mL of streptomycin, and 5 ppm of hemin. Promastigotes were cultured to the stationary phase. The promastigotes’ growth curve was determined by directly counting parasites fixed in 2% paraformaldehyde in trypan blue using a Neubauer chamber.

To screen their activity, all *N*-acylhydrazones were solubilized at 50 mM in DMSO, and 2 µL of each compound was added to 1 mL of promastigote culture (5.0 × 10^5^ cells/mL) to achieve a final concentration of 100 µM. The final DMSO concentration in the culture medium was 0.2%, which did not interfere with parasite growth. The tests were performed using 1.5 mL microtubes. The number of parasites was assessed after 24, 48, and 72 h by direct counting using a Neubauer chamber.

To determine the half-maximal inhibitory concentration (IC_50_), we serially diluted *N*-acylhydrazones two-fold in DMSO to obtain concentrations ranging from 1.25 µM to 100 µM in a final volume of 1 mL of promastigote cultures containing 5 × 10^5^ cells/mL. The treated promastigote cultures were incubated at 25 °C in a biological oxygen demand incubator (B.O.D.) for 72 h. After that, we counted the parasites by mixing 10 µL of the culture with 10 µL of Trypan Blue.

A control experiment was performed by adding DMSO (final concentration 0.2%) or 2 µL of water to the promastigote culture. All experiments were performed in triplicate, and each compound was tested in at least two independent experiments [15].

### 2.4. Arginase Inhibition Test

Recombinant arginase inhibition and screening tests were performed as previously reported [29]. Briefly, the screening of the inhibitory activity of the arginase enzyme was carried out with compounds at 100 µM. The *N*-acylhydrazone derivatives were prepared at 50 mM in DMSO and then dissolved in 50 mM CHES buffer at pH 9.5. A positive control was included without any compound. The assays were performed in triplicate in at least two independent experiments. The activity of arginase was quantified by measuring urea production with the colorimetric method of Berthelot [30]. The absorbance of 200 µL samples was read at 600 nm in an EPOCH2 microplate reader and was used to determine arginase inhibition.

### 2.5. In Silico Drug ADMET and Drug-Likeness Prediction

The assessment of drug absorption, distribution, metabolism, and excretion (ADME) parameters, toxicity prediction, and drug-likeness prediction was performed using the web tools SwissADME [31] and PK-Deep [32].

### 2.6. Statistical Analysis

The data were analyzed and presented as the mean with SEM (standard error of the mean) for N = 6, representing two independent experiments performed in triplicate. The IC_50_ values of compounds **8** and **18** were determined using a nonlinear sigmoidal regression model with a plot of log[Dose] versus growth inhibition. All data were analyzed using GraphPad Prism software (version 8 for Windows, San Diego, CA, USA).

## 3. Results

### 3.1. Chemistry of Compounds

The synthetic route for preparing the hydrazones (**4**–**36**) is shown in Figure 2. The 2-(1H-pyrazol-1-yl)acetohydrazide (**37a**–**d**) could be prepared in 27–86% yield from the reaction of the appropriate pyrazole and ethyl 2-bromoacetate in acetonitrile with K_2_CO_3_ under reflux for 3–24 h and subsequent reaction with hydrazine hydrate. The 2-(1H-pyrazol-1-yl)acetohydrazide **37a**–**d** undergo condensation with aromatic aldehydes in methanol with catalytic HCl to produce the hydrazones (**4**–**36**).

*N*-acylhydrazones (NAH) can be formed as *E*/*Z* diastereomers. However, studies by Palla and collaborators [33,34] showed that NAHs of aromatic aldehydes with benzoylhydrazide are found in solution only in the *E* configuration. This fact is related to the large steric effect caused by the aromatic ring, which, in this isomer, presents less repulsion (Figure 3). X-ray diffraction analyses of NAH crystals of aromatic aldehydes confirm that the preferred diastereomer presents the *E* configuration for the C=N bond of the imine. However, *syn*-periplanar (*sp*) or *anti*periplanar (*ap*) conformations may be present in these crystals depending on the molecular structure in question [35,36,37,38].

In the ^1^H NMR spectrum, NAH **4**–**36** show characteristic signals with chemical shifts between 7.92 and 8.46 ppm for the imino hydrogen (N=CH) and 11.49–12.16 ppm for CONH. However, signal doubling was observed in the ^1^H and ^13^C NMR spectra. Compound **22** was used to exemplify the NMR assignments. Therefore, it was possible to observe a duplicate singlet at 5.26/4.79 ppm for CH2 hydrogens (H8), a duplicate singlet at 2.21/2.16 for CH3 hydrogens (H7), a duplicate singlet at 8.15/8.37 ppm for the imino hydrogen N=CH, and a duplicate singlet at 11.87/11.97 ppm for the CONH hydrogen. In the ^13^C NMR spectrum, duplicate signals were also identified at 10.57/10.49 ppm for the CH3 carbon (C7), as well as at 49.51/50.24 ppm, corresponding to the CH2 carbon (C8), at 104.78/104.66 ppm for C4 carbon of the pyrazole ring, and at 121.14/121.0 ppm, 124.27/124.09 ppm, 130.33/130.29 ppm, 133.13/132.97 ppm, 135.87/135.75 ppm referring, respectively, to carbons C14, C16, C17, C18, and C13 of the aromatic ring, at 146.15/145.8 ppm for C3, at 144.98/141.59 ppm for the imino carbon (N=CH), and at 168.8/163.85 ppm for the C=O carbon (C9) (see Appendix A).

After these experiments, an investigation was carried out to determine whether the duplication of the signals was due to the presence of conformers (*ap* and *sp*) or due to the presence of diastereomers of the *E* and *Z* configurations for NAH **4**–**36**.

Based on this context, two experiments were proposed to evaluate the issue. In the first, ^1^H and ^13^C NMR spectra were performed at a temperature of 75 °C. Therefore, it was possible to note a coalescence of signals in the spectrum when changing the temperature of the experiment, which indicates the presence of *anti*- and *syn*-periplanar conformers of the thermodynamically more stable diastereomer (*E*). In the second experiment, the synthesis of an *N*-methylated derivative (**37**) of NAH **22** was proposed (Figure 3). Thus, the signals of only one of the rotamers should be observed due to the increase in the rotational barrier of the amide bond caused by the presence of the methyl group linked to nitrogen [39]. In the ^1^H and ^13^C NMR spectra of compound **37**, no duplicate signals are observed.

Based on the set of results presented, compounds **4**–**36** were obtained as single (*E*)-diastereomers. In solution, the two stable anti- and syn-periplanar conformers were in equilibrium, interconverted by rotation around the amide bonds. This result is in accordance with our previous work, in which a series of different *N*-acylhydrazones (NAH) was developed [40].

### 3.2. Activity of Compounds Against Leishmania Promastigotes

All 36 compounds were tested against *Leishmania* promastigotes. Compounds **8** and **18** exhibited over 90% inhibition in *Leishmania* cultures. Notably, compounds **8** and **18** were subjected to a 120 h screening period, unlike the other compounds, due to their capacity to inhibit promastigote growth. The remaining compounds showed lower levels of inhibition or no inhibition (Table 1). The half-maximal inhibitory concentration (IC_50_) was determined for compounds **8** and **18**, with values of 10.5 (9.4–11.8) µM and 42.8 (41.3–44.4) µM, respectively. Figure 4 shows the time-dependent action of the compounds, while Figure 5 shows the dose–response curves after 72 h of treatment.

### 3.3. Drug ADMET and Drug-Likeness Prediction

In silico pharmacokinetic properties for compound **8** were assessed using the web tool Deep-PK. The favorable predicted parameters indicate that the compound (MW 331) did not violate Lipinski’s rule of five and suggest good gastrointestinal (GI) absorption, with some caveats regarding blood-brain barrier (BBB) permeability. The Deep-PK toxicity analysis highlighted potential toxicity in the liver and lungs, as well as potential carcinogenicity.

## 4. Discussion

This study provides valuable insights into the study of *N*-acylhydrazone derivatives against *Leishmania amazonensis*, specifically targeting promastigote growth and evaluating the potential for arginase inhibition. Notably, compounds **8** and **18** exhibited significant antipromastigote activity, with inhibition IC_50_ of 10.5 µM and 42.8 µM after 72 h, making them promising candidates for further investigation in leishmaniasis treatment. Interestingly, despite their high antipromastigote efficacy, these compounds showed limited interaction with the arginase enzyme, suggesting that their mechanism of action might not involve direct arginase inhibition, but alternative pathways affecting the parasite’s metabolic or structural integrity.

The exploration of *N*-acylhydrazones as antileishmanial agents aligns with recent efforts to identify compounds with high specificity and low toxicity, addressing the limitations of current leishmaniasis treatments. Many current therapies rely on drugs like pentamidine and amphotericin B, which carry significant side effects and are cost-prohibitive in many endemic regions [41]. The unique chemical properties of *N*-acylhydrazones, particularly their ease of synthesis and stability, make them suitable for further modification and optimization in drug design [42].

The results from the 100 µM assay against promastigotes provide insights into the structure–activity relationship (SAR) of at least six *N*-acylhydrazones evaluated in this study. Firstly, a comparison of compounds **8** (R1 = NO_2_, R2 = Me, Ar = 2,4-dimethyl-Ph) and **27** (R1 = H, R2 = Me, Ar = 2,4-dimethyl-Ph) indicates that the nitro group at position R1 is essential for antileishmanial activity. Similarly, comparing **18** (R1 = H, R2 = Me, Ar = 2-OH-Ph) and **15** (R1 = H, R2 = Me, Ar = 3-OH-Ph) shows that the 3-OH position in Ar = 3-OH-Ph results in a loss of antileishmanial activity. Additionally, comparing **18** (R1 = H, R2 = Me, Ar = 2-OH-Ph) with **23** (R1 = H, R2 = Me, Ar = 2-NO_2_-Ph) shows that the hydroxyl group is responsible for **18**’s superior activity at 100 µM.

Comparing compounds **23** and **36**, which have an identical aryl group, shows that R2 = Me is crucial for activity.

Furthermore, a direct comparison of the IC_50_ values reveals that compound **8** (IC_50_ = 10 µM) is 4.4 times more potent than compound **18** (IC_50_ = 44 µM). These SAR insights could guide the design of novel compounds with enhanced antiparasitic activity for subsequent cytotoxicity investigations.

Given the well-documented activities of pyrazole-based structures in antiparasitic, anti-inflammatory, and anticancer applications, the observed antipromastigote effect could potentially be attributed to membrane depolarization or reactive oxygen species (ROS) production, which could compromise cell membrane integrity [42,43]. This hypothesis is consistent with the known potential of hydrazones to disrupt cellular processes via oxidative stress mechanisms.

Furthermore, the structural flexibility of *N*-acylhydrazone compounds offers a foundation for developing derivatives with enhanced potency and specificity [44]. This study opens avenues for conducting the synthesis of new *N*-acylhydrazones for testing against *Leishmania* and deeper biochemical analyses to elucidate the specific interactions between *N*-acylhydrazones and *Leishmania* spp. Ultimately, these findings contribute to the broader quest for new therapeutic options against leishmaniasis, reinforcing the need for innovative, less toxic, and more affordable treatments for this neglected tropical disease.

## Data Availability

The original contributions presented in this study are included in the Appendix A. Further inquiries can be directed to the corresponding author.

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
