# Peer review of "Design, Synthesis, and Biological Evaluation of N-Acylhydrazones and Their Activity Against Leishmania amazonensis Promastigotes"

_microorganisms, 2025, doi:10.3390/microorganisms13071563_

Round 1

Reviewer 1 Report (Previous Reviewer 3)

Comments and Suggestions for Authors

The authors have satisfactorily addressed many of the reviewer comments in the revised manuscript.   However, the key issue raised by two reviewers regarding the lack of data relevant to the activity of the compounds on intracellular amastigotes, has not been addressed other than to say that since the none of the compounds were active against promastigotes at IC50 values under 10 μM, they were not worth testing further. The authors argue that they intend to use the biological results obtained in this study as a starting point for designing new compounds to reduce IC50 values against promastigotes and minimize toxicity.   Thus, the results of this initial screen must be considered an incremental advance.    

Reviewer 2 Report (Previous Reviewer 2)

Comments and Suggestions for Authors

The authors promptly respond to all of the comments.

Reviewer 3 Report (New Reviewer)

Comments and Suggestions for Authors

Dear Authors, I find your paper interesting and it will be important to know in the future the mechanism of action of these promastigote growth inhibitions and the action against amastigotes.

I only have one objection: in line 41 citation to ref 6 is not necessary.

This manuscript is a resubmission of an earlier submission. The following is a list of the peer review reports and author responses from that submission.

Round 1

Reviewer 1 Report

Comments and Suggestions for Authors

The paper deals with a relevant scientific problem that is finding new alternatives for the treatment of leishmaniasis. However, in our opinion, major changes are needed to achieve the required level for publication in Antimicrobials. A Word version of the manuscript was attached with suggested changes and comments. Major issues found are:

Title: “N-Acylhydrazones Exhibit Potent Activity Against Leishmania amazonensis Promastigotes: Design, Synthesis, and Biological Evaluation”.

1)      None of the compounds had IC50 values under 10 µM which is the minimum cut-off value to consider a compound as a hit (https://doi.org/10.1017/S003118201300142X). From our perspective, a “potent” compound should have an IC50≤ 1 µM.

2)      No elements of “design” were included. Under the term “design”, a reader would expect to find the rationale for the selection of substituents and the expected contribution of the proposed substituents to the molecular properties and/or to the biological activity.

Introduction

No rationale for testing the anti-arginase activity of the test compounds was provided.

Materials and Methods

The description of the “Screening and IC50 of N-acylhydrazone Derivatives in Promastigote Culture” section needs to be improved. Aspects like the following need clarification: 1) Whether tubes or plates (and of what kind) were used; 2) Final concentration of DMSO in the culture medium; 3) Were the compounds serially diluted, what factor of dilution?; 4) Incubation temperature; 5) Volumen of cultures?; 6) Volume of samples taken at each time-point.

The method used to assess the anti-promastigote activity, although valid, is not the most efficient. There are many recent methods that objectively and efficiently estimate the number of promastigotes which are cheap and require minimum equipment.

Promastigotes are not the relevant parasite stage, since they are not present in humans. Therefore, intracellular amastigotes are the preferred stage for the screening of antileishmanial compounds. Moreover, considering that Leishmania amastigotes reside intracellularly, specifically inside parasitophorous vacuoles, an antileishmanial compound should be able to permeate the host cell membrane, the parasitophorous vacuole membrane and in many cases, the amastigote membrane. Additionally, that should occur without damaging the host (mammalian) cell. Consequently, to ascertain that a compound has selective antileishmanial activity it should be tested against intracellular amastigotes and the cytotoxicity for the host cell should be tested in parallel. IC50 against intracellular amastigotes should be under 10 µM and the selectivity index (50 % Cytotoxic concentration/IC50) should be over 10. In conclusion, in the present work, the method used to assess the antileishmanial activity of the test compounds was not appropriate.

Results and discussion

An explanation of why the arginase inhibitory activity of the test compounds was evaluated and then an analysis of why none of them were active.

A discussion was included regarding how the present work could contribute to the identification of compounds with high specificity and low toxicity compared to current antileishmanial drugs; however, the present work did not include any test of toxicity, either in vitro on in vivo.

Others

Use of the International System of Units should be followed.

Reviewer 2 Report

Comments and Suggestions for Authors

I have read the submitted manuscript titled" N-Acylhydrazones Exhibit Potent Activity Against Leishmania 2 amazonensis Promastigotes: Design, Synthesis, and Biological 3 Evaluation". The authors conclude N-acylhydrazone derivatives as a potential treatment for leishmaniasis. This class of compounds has been investigated for various therapeutic applications, including antiparasitic agents. These compounds have great potential for chemical and pharmaceutical modifications.  The authors have done an excellent job of presenting the experimental data. However, a few specific points should be addressed to enhance the quality and precision of the work.

The major flaws of this report are:

-          The introduction part requires substantial writing and rewording to provide more information regarding acylhydrazones

-          Consider revising your material and procedures, as well as your study's in vivo design

-          Separate the "Results" and "Discussion" parts and rework them considerably.

-          The manuscript should be extensively updated to enhance punctuation, grammar, and readability.

Minor flaws

Line 12: "Leishmaniasis, a significant" replace with "Leishmaniasis is a significant"

Line 13 & line 92: "this research" replace with 'this study'

Line 75:& 587 ' in vivo" replace with " in vivo"

Line 107: Sigma–Aldrich, ……..." add country name"

Line 460: IC50 replace with "IC50"

Reviewer 3 Report

Comments and Suggestions for Authors

Prior studies have shown the in vitro activity of some pyrazole derivatives against Leishmania.    In the current studies a series of pyrazole acylhydrazones was designed to evaluate the activity against L. amazonensis promastigotes in vitro.    The advantage of these compounds, in addition to their anti-microbial potential, is that they are easily synthesized and highly stable.    Of the 33 novel acylhydrazone-derived compounds that were screened, two were found to show strong inhibition of promastigote growth (IC50 of 10.5 µM and 42.8 µM).    The expected inhibitory activity of the compounds against arginase was not observed, and the mechanism of action is not known.     The design, synthesis and analysis of the various hydrazone compounds is well described.     The main deficit of the paper is the lack of screening against intracellular amastigotes, as this parasite stage is the only therapeutic target and activity against amastigotes cannot be assumed based on the effect observed in promastigote cultures.  At a minimum, the two active compounds should have been tested in vitro using infected macrophages, either macrophage cell lines or primary explant cultures of blood monocyte-derived macrophages (human), or bone marrow or peritoneal macrophages (mouse).   

In the text, it is stated that compounds 8 and 18 were subjected to a 120-hour screening period, but nowhere is this data shown.   

It is important to distinguish between the leishmanistatic and leishmanicidal activities of the drugs.   The way the data in figs 4&5 are calculated as a comparison with the control culture, it is not possible to know whether the inhibitory effect of the compounds is explained solely by slower promastigote growth, or if there is also leishmanicidal activity.        Do parasite numbers decline during 120 hr culture with either compound 8 or 18?  Are dead promastigotes observed in the cultures?